# Multimodal Physical Adversarial Clothing Evades Visible-Thermal Detectors with Non-Overlapping RGB-T Pattern

## Abstract

Visible-thermal (RGB-T) object detection is a crucial technology for applications such as autonomous driving, where multimodal fusion enhances performance in challenging conditions like low light. However, the security of RGB-T detectors, particularly in the physical world, has been largely overlooked. This paper proposes a novel approach to RGB-T physical attacks using adversarial clothing with a non-overlapping RGB-T pattern (NORP). To simulate full-view (0°–360°) RGB-T attacks, we construct 3D RGB-T models for human and adversarial clothing. NORP is a new adversarial pattern design using distinct visible and thermal materials without overlap, avoiding the light reduction in overlapping RGB-T patterns (ORP). To optimize the NORP on adversarial clothing, we propose a spatial discrete-continuous optimization (SDCO) method. We systematically evaluated our method on RGB-T detectors with different fusion architectures, demonstrating high attack success rates both in the digital and physical worlds. Additionally, we introduce a fusion-stage ensemble method that enhances the transferability of adversarial attacks across unseen RGB-T detectors with different fusion architectures.

## 1 Introduction

Visible-thermal (RGB-T) object detection is a type of multimodal object detection with important applications in fields such as autonomous driving and medical AI (Nawaz et al., 2025; Lai et al., 2024; Dasgupta et al., 2022). For example, in challenging conditions such as adverse weather or nighttime, RGB-T object detection leverages thermal imaging to compensate for the performance degradation of visible-only detectors while producing clearer predictions than thermal-only detectors by preserving more details from visible-light images. This advantage significantly enhances the robustness of autonomous driving systems across diverse scenarios. Based on different fusion strategies for multimodal information, RGB-T object detectors can be classified into four categories: early-fusion, mid-fusion, late-fusion, and independent visible and thermal detectors.

Despite the widespread applications of RGB-T detectors, their security has received little attention because multimodal detectors are commonly assumed to be relatively robust. However, this is very important for the safety of AI systems in real-world applications, such as autonomous driving. To address this issue, physical adversarial examples (Wei et al., 2024) offer an effective approach to identifying vulnerabilities in AI systems deployed in the physical world and inspiring novel defense strategies. Currently, most physical adversarial examples focuses either solely on the visible modality (Thys et al., 2019; Xu et al., 2020; Wu et al., 2020; Hu et al., 2021; 2022; 2023; Wei et al., 2022) or the thermal modality (Zhu et al., 2021; 2022; 2024; Wei et al., 2023a;c). Due to the significant differences in imaging mechanisms between these two modalities, adversarial examples crafted for one modality cannot be effectively transferred to the other. As a result, these physical adversarial examples lack the ability to attack multimodal detectors.

To the best of our knowledge, only three methods have been proposed for physical adversarial attacks in visible-thermal settings, including MAP (Kim et al., 2022), UAP (Wei et al., 2023b), and MIC (Kim et al., 2023). However, they have two major limitations. First, MAP and UAP are realized as 2D patches, which can only attack detectors at a narrow range of viewing angles (e.g., -30° to 30°).

Second, MIC deploys an overlapping RGB-T pattern (ORP) by attaching multiple special low-E films onto a printed fabric, which diminishes the visibility of the printed adversarial pattern and increases production costs. These limitations result in the vulnerability of RGB-T detectors across different physical settings not being fully explored.

To address the first limitation, we construct aligned 3D RGB-T models of the human body and clothing to simulate full-view (0°–360°) attacks in the digital world, and manufacture corresponding 3D RGB-T adversarial clothing to enable full-view attacks in the physical world. To address the second limitation, we propose a non-overlapping RGB-T pattern (NORP) design for RGB-T adversarial clothing. NORP not only leverages the visible and thermal properties of different materials simultaneously but also ensures that these materials do not overlap, thereby avoiding the light reduction problem inherent to ORP. Moreover, we use commonly available materials (fabric and aluminum film) to deploy NORP, which offers the advantage of low cost.

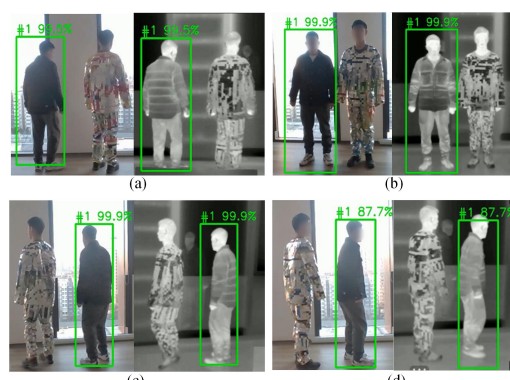

Figure 1: **Demonstration of physical attacks against RGB-T detectors with different fusion architectures.** (a) early-fusion detector, (b) mid-fusion detector, (c) late-fusion detector, and (d) independent RGB-T detectors at multiple views. The individual wearing adversarial clothes remained undetected in both modalities, while the individual wearing ordinary clothes was detected (indicated by a bounding box). See *Supplementary Material* for the *Demo Video*.

A key challenge of optimizing NORP lies in the spatial dependency between the optimization variables of the visible and thermal patterns. For example, when a pixel is chosen to be aluminum film, its RGB values are fixed and cannot be simultaneously optimized as continuous variables, and vice versa. To address this challenge, we propose a spatial discrete-continuous optimization (SDCO) method that enables simultaneous optimization of continuous RGB pixels and discrete thermal pixels within NORP.

Different from previous methods (Kim et al., 2022; Wei et al., 2023b; Kim et al., 2023) that validate on only one type of (e.g., mid-fusion) RGB-T detector, we systematically evaluate our approach on RGB-T detectors with various fusion architectures, both in the digital and physical world. Experiments show that our RGB-T adversarial clothes effectively attacked RGB-T detectors across multiple fusion architectures, with average ASR of 99.6% and 71.0% in the digital and physical worlds, respectively. Fig. 1 shows several visualized examples. Furthermore, we propose a fusion-stage ensemble method that enables a piece of adversarial clothing to simultaneously attack RGB-T detectors with multiple fusion architectures. Our work comprehensively reveals the vulnerabilities of RGB-T detectors across different fusion architectures, viewing angles, and distances, which is important for building more robust detectors in the future.

## 2 RELATED WORK

### 2.1 RGB-T OBJECT DETECTION

RGB-T object detectors are multimodal object detectors that integrate visible-light and thermal imaging modalities.(El Ahmar et al., 2023; Zhang et al., 2024) which integrate multimodal information at the image level, mid-fusion detectors (Chen et al., 2022; Guo et al., 2024) which focus on feature-level fusion, late-fusion detectors (Liu et al., 2016; Ni et al., 2022) which merge multimodal information at the prediction stage, and independent visible and thermal detectors (Khanam & Hussain, 2024; Zhu et al., 2020),

where each modality operates independently.

### 2.2 PHYSICAL ATTACKS IN VISIBLE-THERMAL MODALITY

To the best of our knowledge, only a limited number of works focus on visible-thermal physical attacks. Such works require not only the simultaneous optimization of visible and thermal variables

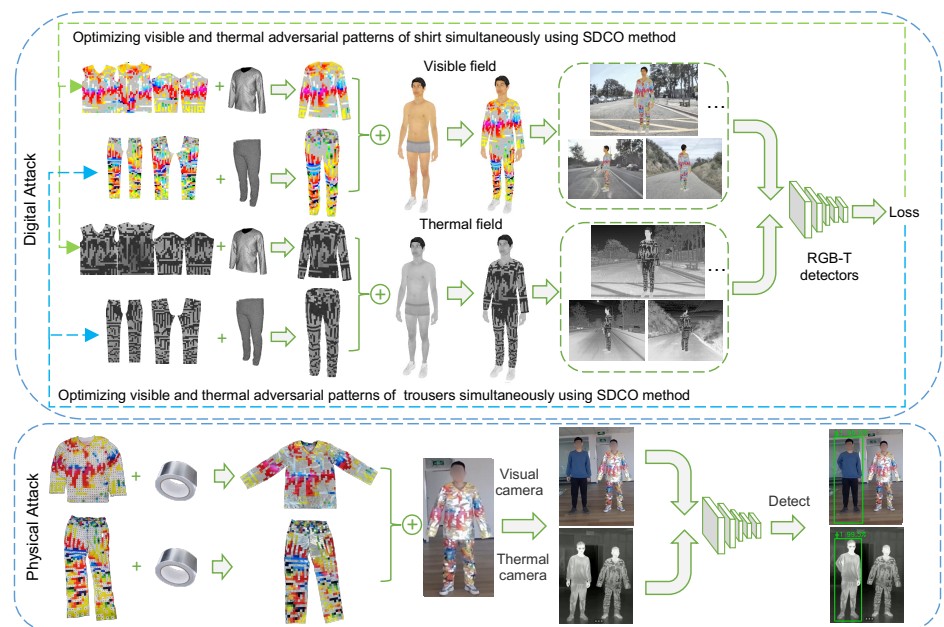

Figure 2: The overall pipeline of the proposed method.

but also the physical implementation of both modalities. Kim et al. (Kim et al., 2022) proposed a multispectral adversarial patch (MAP) to attack a mid-fusion RGB-T person detector. Later, they (Kim et al., 2023) introduced a low-E film–based clothing, named MIC, to attack the same RGB-T detector. Wei et al. (Wei et al., 2023b) developed a unified adversarial patch (UAP) to evade independent RGB-T detectors. However, due to their limitations either in attack angles or physical settings, the vulnerability of RGB-T detectors across different physical settings has not been fully explored.

## 3 METHOD

### 3.1 PROBLEM FORMULATION

For a given target $x$, its visible image is denoted as $x_{\text{vis}}$, while its thermal image is denoted as $x_{\text{thm}}$. After adding adversarial perturbations, the adversarial images in both modalities are represented as $x_{\text{vis}}^{\text{adv}}$ and $x_{\text{thm}}^{\text{adv}}$, respectively. The objective of an RGB-T attack is to ensure that the RGB-T detector $f$ fails to detect the target $x$. Given a detection threshold $q$, the optimization objective is:

$$f(x_{\text{vis}}^{\text{adv}}, x_{\text{thm}}^{\text{adv}}) < q. \tag{1}$$

Our RGB-T attack pipeline is shown in Fig. 2. First, we construct aligned 3D RGB-T models for both the human body and clothing. Then we design the non-overlapping RGB-T pattern (NORP) and apply our SDCO method to optimize the visible and thermal adversarial patterns simultaneously. Finally, based on the optimization results, we manufacture the multimodal physical clothes and evaluate their performance in the physical world.

### 3.2 BUILDING 3D RGB-T MODELS

To simulate full-view RGB-T attacks, we need to construct an aligned 3D RGB-T model. Since most existing 3D models are RGB models, we develop a method for extending an 3D RGB model into an aligned 3D RGB-T model based on a previous thermal 3D modeling approach (Zhu et al., 2024). Initially, we utilized publicly available 3D RGB human and clothing models (Hu et al., 2023) as the foundation. Next, we take a clothing model as an example to illustrate our method.

The challenge in building an aligned 3D RGB-T model lies in generating an thermal "skin" that aligns with the 3D mesh model. To address this, we first unfold the faces of the 3D mesh model into a 2D faces map and organize it into different regions, such as the back and arms, using Maya software. Next, we capture real thermal images of clothing using an thermal camera and process

them to align with the faces map, producing an aligned thermal texture map. See App. A for how these photos were captured and processed. This process ensures that the real thermal texture is properly aligned with the 3D mesh model.

## 3.3 Designing Non-Overlapping RGB-T Pattern

After constructing the 3D model of the clothing, we propose to design non-overlapping RGB-T pattern (NORP) for the multimodal adversarial clothing. Since we use printable fabric and aluminum film to deploy NORP onto adversarial clothing, each location on the clothing is either printed with RGB colors or covered with aluminum film, but not both. We parameterize the NORP, so that it can be represented by $N$ pixels. Let $X = [X_i] = [r_i, g_i, b_i, t_i]_{i=1,2,\ldots,N}$ represent the NORP, where $[r_i, g_i, b_i]$ denotes the RGB value of visible light and $t_i$ denotes the thermal emission intensity.

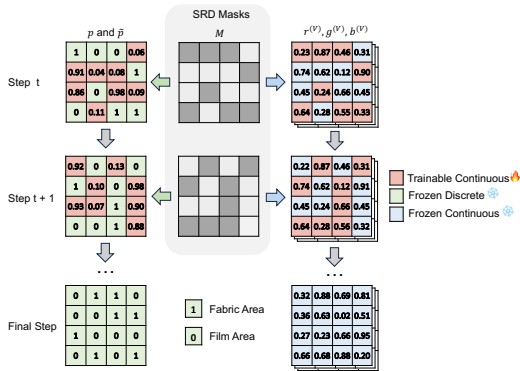

Figure 3: **Illustration of the SDCO method.** In SRD, black pixels represent discretizing $\tilde{p}$ and updating $r^{(V)}, g^{(V)}, b^{(V)}$, while gray pixels represent freezing $r^{(V)}, g^{(V)}, b^{(V)}$ and updating $\tilde{p}$. At final step, all $\tilde{p}$ are discretized, and NORP optimization is finished.

If the clothing at the pixel $i$ is covered with an aluminum film, its RGB-T value is fixed by $[r^{(T)}, g^{(T)}, b^{(T)}, t^{(\mathrm{film})}]$, which is determined by the characteristic of the aluminum film. Otherwise, the fabric can be printed with any RGB color, and the body temperature determines the thermal intensity. Therefore, its RGB-T value can be represented by $[r_i^{(V)}, g_i^{(V)}, b_i^{(V)}, t_i^{(\mathrm{body})}]$, where $t_i^{(\mathrm{body})}$ is sampled from the thermal texture. Furthermore, we use an additional variable $p_i = 0$ to represent that a pixel $i$ is covered with an aluminum film, and $p_i = 1$ otherwise. The RGB-T pixel $X_i = [r_i, g_i, b_i, t_i]$ thus can be represented by

$$X_i = \mathrm{H}(Y_i) = p_i \cdot [r_i^{(V)}, g_i^{(V)}, b_i^{(V)}, t_i^{(\mathrm{body})}] + (1 - p_i) \cdot [r^{(T)}, g^{(T)}, b^{(T)}, t^{(\mathrm{film})}], \qquad (2)$$

where $Y_i = [r_i^{(V)}, g_i^{(V)}, b_i^{(V)}, p_i]$ is a collection of learnable variables. Please note that the values of $r^{(T)}, g^{(T)}, b^{(T)}, t_i^{(\mathrm{body})}, t^{(\mathrm{film})}$ are measured values (constants).

## 3.4 Optimizing the Adversarial RGB-T Pattern

We develop a gradient-based algorithm to optimize the variables of NORP. Since $p_i$ is discrete in 2, we optimize a continuous variable $\tilde{p}_i$ instead, and discretize $\tilde{p}_i$ as $p_i = \Bbbk(\tilde{p}_i \geq 0.5)$ after the optimization. However, a naive method to optimize the variables $[r_i^{(V)}, g_i^{(V)}, b_i^{(V)}, \tilde{p}_i]$ directly led to suboptimal results (see 4.6 for details). This is because the continuous variables $r_i^{(V)}, g_i^{(V)}, b_i^{(V)}$ and the discrete variable $p_i$ are entangled, thus the approximation of $p_i$ affects other variables as well.

Therefore, we propose a spatial discrete-continuous optimization (SDCO) method based on Spatially-Random Discretization (SRD). The core idea of SRD is to discretize some pixels during the gradient optimization process, while computing gradients for other pixels that remain continuous. As shown in Fig. 3, during each iteration, we apply a random mask to discretize a portion of the thermal adversarial pattern's pixels and freeze these variables. The random discretization ratio is denoted as $\alpha$, with the corresponding visible variables in these regions being optimized. Conversely, the remaining $1 - \alpha$ portion of the thermal adversarial pattern's pixels is set to be trainable, while the corresponding visible pixels are frozen. As the mask changes randomly throughout the iterations, each pixel has an equal probability of being in the trainable state, allowing for iterative updates.

This method balances the optimization of visible and thermal adversarial patterns while satisfying the spatial interdependency constraints between these two patterns specified in Equation 2, ensuring that the multimodal adversarial pattern is physically realizable. The process of the algorithm is outlined in Algorithm 1.

---

**Algorithm 1** Spatial Discrete-Continuous Optimization

---

**Require:** Variables $Y = \{ [r_i^{(\mathrm{V})}, g_i^{(\mathrm{V})}, b_i^{(\mathrm{V})}, p_i] \}_{i=1}^N$, continuous $\tilde{p}_i$, step size $\eta$, discretization probability $\alpha$, max iterations $T$, detection process $f(X)$ in Equation 1, function $\mathrm{H}(Y)$ in Equation 2, loss $\mathcal{L}$ in Equation 7
 1: Initialize $j \leftarrow 0$
 2: **while** $j < T$ **do**
 3:     Generate random mask $M \in \{0,1\}^N$ with $M_i \sim \mathrm{Bernoulli}(\alpha)$       *% Spatially-Random Discretization*
 4:     **for** each pixel $i$ **do**
 5:        $p_i \leftarrow \mathbb{I}(\tilde{p}_i \geq 0.5)$ if $M_i = 1$ else $p_i \leftarrow \tilde{p}_i$ *% Discretize thermal / keep thermal continuous*
 6:     **end for**
 7:     $L \leftarrow \mathcal{L}\big(f(\mathrm{H}(Y))\big)$                      *% One single forward & backward step*
 8:     $\nabla_Y L \leftarrow \frac{\partial L}{\partial Y}$
 9:     **for** each pixel $i$ **do**
10:        Set $\nabla_{p_i} L \leftarrow 0$ if $M_i = 1$ else $\nabla_{[r_i^{(\mathrm{V})}, g_i^{(\mathrm{V})}, b_i^{(\mathrm{V})}]} L \leftarrow 0$    *% Block thermal vs. RGB gradient*
11:     **end for**
12:     Update $Y \leftarrow Y - \eta \nabla_Y L$; $j \leftarrow j + 1$
13: **end while**
14: **Final Binarization:** $p_i \leftarrow \mathbb{I}(\tilde{p}_i \geq 0.5)$ for all $i$;
15: **return** Adversarial Texture $X = \mathrm{H}(Y)$

---

### 3.5 Applying the Adversarial RGB-T Patterns to 3D RGB-T Models

Let $T_{\mathrm{adv}}^{\mathrm{vis}}$ and $T_{\mathrm{adv}}^{\mathrm{thm}}$ denote the optimized adversarial textures for the visible and thermal modalities, respectively. We first apply EOT algorithm (Athalye et al., 2018) to simulate physical perturbations during optimization. The adversarial textures after the EOT transformation are represented as:

$$T_{\mathrm{adv\text{-}E}}^{\mathrm{vis}} = \mathrm{EOT}\big(T_{\mathrm{adv}}^{\mathrm{vis}}\big), \quad T_{\mathrm{adv\text{-}E}}^{\mathrm{thm}} = \mathrm{EOT}\big(T_{\mathrm{adv}}^{\mathrm{thm}}\big). \tag{3}$$

We then use the renderer $\mathrm{R}$ to map the adversarial textures onto the surface of the 3D clothing mesh $M_{\mathrm{cloth}}$. The parameters $\phi$ of the renderer include rendering distances, angles, etc. The rendered adversarial clothing images in both visible and thermal modalities are represented as:

$$I_{\mathrm{cloth}}^{\mathrm{vis}} = \mathrm{R}(M_{\mathrm{cloth}}, T_{\mathrm{adv\text{-}E}}^{\mathrm{vis}}, \phi), \quad I_{\mathrm{cloth}}^{\mathrm{thm}} = \mathrm{R}(M_{\mathrm{cloth}}, T_{\mathrm{adv\text{-}E}}^{\mathrm{thm}}, \phi). \tag{4}$$

Next, we combine the 3D RGB-T person models and the 3D RGB-T clothing models. In other words, we let the 3D person "wear" the 3D clothing. The 3D RGB-T person model consists of the 3D body model $M_{\mathrm{body}}$, the RGB skin images $P_{\mathrm{skin}}^{\mathrm{vis}}$, and the thermal skin image $P_{\mathrm{skin}}^{\mathrm{inf}}$. Therefore, the rendered visible and thermal images of a person wearing the clothing are given by:

$$I_{\mathrm{person}}^{\mathrm{vis}} = \mathrm{R}(M_{\mathrm{body}}, P_{\mathrm{skin}}^{\mathrm{vis}}, I_{\mathrm{cloth}}^{\mathrm{vis}}), \quad I_{\mathrm{person}}^{\mathrm{thm}} = \mathrm{R}(M_{\mathrm{body}}, P_{\mathrm{skin}}^{\mathrm{thm}}, I_{\mathrm{cloth}}^{\mathrm{thm}}). \tag{5}$$

To simulate physical attacks under different environments, we paste the rendered images $I_{\mathrm{person}}^{\mathrm{vis}}$ and $I_{\mathrm{man}}^{\mathrm{thm}}$ onto the aligned visible and thermal background images $I_{\mathrm{back}}^{\mathrm{vis}}$ and $I_{\mathrm{back}}^{\mathrm{thm}}$, respectively. The pasted images are given by:

$$I_{\mathrm{paste}}^{\mathrm{vis}} = \mathrm{Paste}(I_{\mathrm{person}}^{\mathrm{vis}}, I_{\mathrm{back}}^{\mathrm{vis}}), \quad I_{\mathrm{paste}}^{\mathrm{thm}} = \mathrm{Paste}(I_{\mathrm{person}}^{\mathrm{thm}}, I_{\mathrm{back}}^{\mathrm{thm}}). \tag{6}$$

### 3.6 Loss Functions

We input the pasted images $I_{\mathrm{paste}}^{\mathrm{vis}}$ and $I_{\mathrm{paste}}^{\mathrm{thm}}$ into the RGB-T detection model $f$. The optimization objective of the adversarial pattern is to minimize the RGB-T detector's confidence score $f_{\mathrm{obj}}$ of the person wearing the adversarial clothing. Specifically, the optimization loss function is given by:

$$L = f_{\mathrm{obj}}(I_{\mathrm{paste}}^{\mathrm{vis}}, I_{\mathrm{paste}}^{\mathrm{thm}}). \tag{7}$$

To improve the transferability of the adversarial pattern across different fusion architectures of RGB-T detectors, we propose a fusion-stage ensemble method. This approach integrates multiple RGB-T

detectors with different fusion architectures, including early-fusion, mid-fusion, late-fusion, and independent RGB-T detectors during optimization. The ensemble optimization loss function can be formulated as follows:

$$L_{\text{ensemble}} = w_1 \cdot L_{\text{early}} + w_2 \cdot L_{\text{mid}} + w_3 \cdot L_{\text{late}} + w_4 \cdot L_{\text{indep}}. \tag{8}$$

where $w_i, i = 1, 2, 3, 4$ are the empirically determined weights for each fusion stage.

### 3.7 PHYSICAL IMPLEMENTATION

After obtaining the optimized adversarial textures using SDCO method, we print the adversarial textures with RGB color onto fabric. For regions requiring the pasting of aluminum film, the fabric was marked with an "X" to indicate their placement. Each pixel on the fabric measures 25mm × 25mm. A tailor then processed the printed fabric into clothing, including a shirt and pants, and pasted the aluminum film (only 0.1 mm thick) into the clothing regions marked with "X". Through this process, we successfully created physical RGB-T adversarial clothing.

## 4 EXPERIMENTS

### 4.1 DATASET

To simulate RGB-T attacks in various real-world environments during simulation, we used an aligned RGB-T dataset (Zhang et al., 2020) named FLIR-aligned. It contains 4,129 well-aligned RGB-T image pairs for training and 1,013 RGB-T image pairs for testing. We used this dataset as the background images in our 3D RGB-T simulations.

### 4.2 TARGET RGB-T DETECTORS

We selected typical RGB-T detectors across different fusion architectures as our primary attack targets, including an early-fusion detector Prob-E (Chen et al., 2022), a mid-fusion detector Prob-M (Chen et al., 2022), a late-fusion detector Prob-L (Chen et al., 2022), and independent RGB-T detectors YOLOv11(RGB) and YOLOv11(T) (Khanam & Hussain, 2024). After performing attacks on these RGB-T detectors in a white-box setting, we transferred our method to unseen black-box RGB-T detectors to evaluate its transferability. These black-box detectors include an early-fusion detector RPN-E (Liu et al., 2016), a mid-fusion detector AR-CNN (Zhang et al., 2019), a late-fusion detector RPN-L (Liu et al., 2016), and independent RGB-T detectors D-DETR(RGB) and D-DETR(T) (Zhu et al., 2020). All these detectors achieve high detection confidence scores (above 0.85) on clean RGB-T pedestrian targets, making them strong baselines for evaluating adversarial robustness.

### 4.3 EVALUATION METRICS

In our experiments, we adopted the widely used Attack Success Rate (ASR) as the evaluation metric. ASR is defined as the ratio of the number of undetected target pedestrians to the total number of target pedestrians. Similar to previous works (Wei et al., 2022; 2023b; Zhu et al., 2021; 2022; 2024), we set the Intersection over Union (IoU) threshold between the predicted bounding box and the ground truth bounding box to 0.5, and the confidence threshold

Table 1: Comparison of different methods in digital world.

| Method | ASR (%) ↑ | | | | |
|---|---|---|---|---|---|
| | Prob-E | Prob-M | Prob-L | YOLOv11 | |
| | | | | RGB | $T$ |
| Clean | 0.2 | 0.4 | 0.2 | 0.4 | 0.2 |
| Random | 15.6 | 12.0 | 3.4 | 0.2 | 0.6 |
| MAP | 31.4 | 37.2 | 11.2 | 6.8 | 4.2 |
| MIC | 26.2 | 24.0 | 12.4 | 5.8 | 4.0 |
| UAP | 25.4 | 27.8 | 5.6 | 2.8 | 4.4 |
| Ours | **100.0** | **100.0** | **99.8** | **98.8** | **99.4** |

to 0.6. The reported ASR was computed as the average success rate across different viewing angles, distances, and scene conditions.

### 4.4 ATTACK RGB-T DETECTORS IN THE DIGITAL WORLD

Based on the method introduced in Sec. 3.3, we designed the non-overlapping adversarial RGB-T patterns where each pixel in the pattern measures 10 × 10 (see App. D for further analysis of this parameter). Next, we optimized the adversarial patterns using the SDCO method presented in Sec.

3.4, with the mask probability $\alpha$ for SRD set to 70%. A further analysis of $\alpha$ can be found in Sec. 4.6. Additional experimental settings are detailed in App. B. After optimization, we obtained the adversarial clothing textures for RGB-T detectors with different fusion architectures, along with the rendered 3D RGB-T human models with adversarial clothing (Fig. 2). See App. 8 for the optimized 3D RGB-T models for different RGB-T detectors.

Next, we applied the rendered 3D RGB-T human model to the test set of the FLIR-aligned dataset, following the process described in Sec. 3.5. The generated images were then input into RGB-T detectors with different fusion architectures including Prob-E, Prob-M, Prob-L, and YOLOv11 to evaluate the attack effectiveness of our adversarial pattern in the digital world. For a fair comparison, we employed the clean clothing pattern (pure color pattern), random RGB-T pattern (without optimization), and the adversarial patterns generated by previous works including MAP, UAP, and MIC. These patterns were applied according to their original papers and released codes. We used ASR as the evaluation metric. The results are shown in Tab. 1. Our method achieved an average ASR of 99.6% across RGB-T detectors with different fusion stages, while the ASR for the control group was below 37.2%. A set of visual examples comparing these methods is shown in App. 9.

This indicates that our method effectively attacked RGB-T detectors with different fusion architectures in the digital world, outperforming simple baselines and previous RGB-T attack methods.

### 4.5 ANALYSIS OF ADVERSARIAL EFFECTS AT DIFFERENT DISTANCES AND ANGLES

We further analyzed the ASRs of our method at various viewing angles and distances. The viewing angle varied from 0 to 360 degrees, with samples taken every 18 degrees. The distance ranged from 2.5 meters to 20 meters, with samples taken every 2.5 meters. Fig. 4 shows the analysis of ASRs for RGB-T detectors with different fusion architectures across various distances and angles.

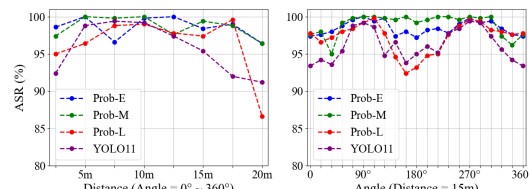

Figure 4: The attack **success** rate for different RGB-T detectors at various (a) distance and (b) viewing angles.

The results show that our method successfully attacked RGB-T detectors with various fusion architectures at different (full-angle) viewing angles and distances. In comparison, some previous methods (Kim et al., 2022; Wei et al., 2023b) are only effective within a limited range of angles and distances (e.g., angles from -30 to 30 degrees, distances between 3 to 6 meters). This can be attributed to the fact that our 3D modeling-based approach can simulate attacks over a broader range of angles (full view) and distances, which is a significant improvement over traditional 2D simulation-based methods.

### 4.6 ABLATION STUDY FOR SDCO METHOD

The core technique of SDCO is Spatially-Random Discretization (SRD). We conducted ablation experiments to evaluate the effectiveness of SRD while keeping all other experimental settings the same as in Sec. 4.4. The results, shown in Tab. 2, indicate that SRD effectively improved the attack ef-

Table 2: Ablation study for SDCO method.

| Method | ASR (%) ↑ | | | | |
|---|---|---|---|---|---|
| | **Prob-E** | **Prob-M** | **Prob-L** | **YOLOv11** | |
| | | | | RGB | $T$ |
| w/o SRD | 78.6 | 88.4 | 67.2 | 48.2 | 46.4 |
| w SRD | **100.0** | **100.0** | **99.8** | **98.8** | **99.4** |

fectiveness against RGB-T detectors with different fusion architectures. This is because SRD effectively balances the dual-modality variable optimization.

We further analyzed the impact of the key parameter $\alpha$ of SRD on the ASR, which serves as a balancing factor between the optimization parameters of the visible and thermal modalities. The results are detailed in C. We observed that the highest average ASR was achieved when $\alpha$ was set to 0.7. Therefore, we set $\alpha = 0.7$ in our experiments.

### 4.7 COMPARISION WITH OTHER OPTIMIZATION METHODS

For simultaneously optimizing discrete and continuous variables, there are other optimization algorithms such as Gumbel-Softmax (Jang et al., 2017) and Straight-Through Estimator (Bengio et al.,

2013) (STE) . Gumbel-Softmax outputs a soft distribution during training and uses a hard distribution during inference, while STE outputs a hard distribution during the forward process and uses a differentiable approximation during the backward process. Both methods transform the discrete optimization problem into an approximate continuous optimization problem, enabling the use of gradient-based methods to optimize both types of variables. We applied these methods to optimize adversarial textures while keeping the settings consistent with Sec. 4.4.

The experimental results, shown in Tab. 3, indicates that in our experiments, our SDCO method outperforms Gumbel-Softmax and STE. This is likely because Gumbel-Softmax and STE essentially perform continuous optimization and discrete operations in different sequential stage (e.g., training vs. inference, forward vs. backward),

Table 3: Comparison with other optimization methods

| Method | ASR (%) ↑ | | | | |
| --- | --- | --- | --- | --- | --- |
| | Prob-E | Prob-M | Prob-L | YOLOv11 | |
| | | | | RGB | $T$ |
| Gumbel | 78.6 | 87.8 | 60.0 | 34.0 | 22.6 |
| STE | 95.6 | 96.8 | 94.4 | 92.4 | 86.8 |
| SDCO | **100.0** | **100.0** | **99.8** | **98.8** | **99.4** |

while our method simultaneously performs continuous optimization and discrete operations in different spatial aeras, which aligns with the spatial distribution of two types of variables in our RGB-T adversarial clothing.

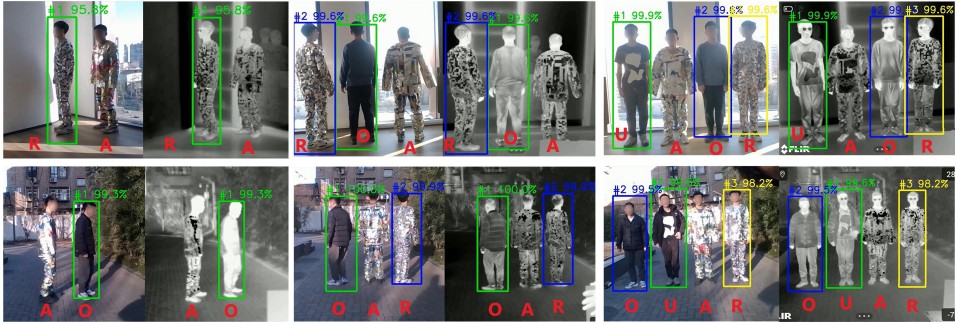

Figure 5: Visualization of physical RGB-T attacks across diverse scenes. Top row: indoor scenarios. Bottom row: outdoor scenarios. O: ordinary clothes. R: random pattern clothes. U: UAP patch. A: adversarial clothes. See *Supplementary Material* for the *Demo Video* and more examples.

## 4.8 ATTACK RGB-T DETECTORS IN THE PHYSICAL WORLD

Based on the results of 3D digital simulation, we created physical RGB-T adversarial clothes according to the process described in Sec. 3.7. To ensure a fair comparison, we selected ordinary clothing, random RGB-T pattern clothing, latest RGB-T attack method UAP patch as control groups (MAP and MIC were excluded due to the lack of corresponding special materials such as low-E films, which makes

Table 4: Comparison of different methods in the physical world. Highest ASR values are marked in **bold**.

| Method | ASR (%) ↑ | | | | |
| --- | --- | --- | --- | --- | --- |
| | Prob-E | Prob-M | Prob-L | YOLOv11 | |
| | | | | RGB | $T$ |
| Clean | 15.2 | 19.6 | 15.3 | 9.4 | 11.6 |
| Random | 23.4 | 21.5 | 17.6 | 11.0 | 9.7 |
| UAP | 15.6 | 23.3 | 15.3 | 8.8 | 12.2 |
| Ours | **73.5** | **76.5** | **79.2** | **61.2** | **64.4** |

physical reproduction difficult). See App. 12 for the photos of our adversarial clothes and the control groups.

We tested the physical-world attack effectiveness of RGB-T adversarial clothes and the control groups. We invited 5 volunteers to participate in the experiments. The human-related experiments have been approved by the Institutional Review Board (IRB). Volunteers participated in groups of two, three, or four, wearing our RGB-T adversarial clothing, random-pattern RGB-T clothing, ordinary clothing, or holding a UAP patch. We used an iPhone 13 Pro and a FLIR T560 thermal camera to simultaneously capture visible and thermal images of these volunteers. The captured scenes included both indoor and outdoor environments, spanning different times of the day, includ-

ing morning, noon, afternoon, and nightfall. The camera angles covered a full 0-360° range, and the capture distances ranged from 2 to 15 meters. We recorded 116 videos and sampled 5220 visible-thermal image pairs at a frame rate of 1 frame per second. The collected images were then input into the corresponding RGB-T detectors for detection, and the ASR was calculated, as shown in Tab. 4. Some examples are shown in Fig. 5. See *Supplementary Material* for the *Demo Video*.

The results indicate that our adversarial clothing successfully evades multiple RGB-T detectors with different fusion architectures across various scenes, angles, and distances, consistently outperforming the baseline methods.

### 4.9 ATTACK TRANSFERABILITY

Table 5: Transferability in the digital world. The numbers are ASRs. See App. F for the physical-world results.

| Train \ Test | Prob-E | Prob-M | Prob-L | YOLOv11 | RPN-E | AR-CNN | RPN-L | D-DETR |
|---|---|---|---|---|---|---|---|---|
| Prob-E | 100.0 | 99.0 | 11.2 | 1.0 | 95.4 | 67.8 | 96.2 | 48.6 |
| Prob-M | 81.8 | 100.0 | 39.0 | 0.4 | 92.4 | 64.4 | 92.1 | 70.6 |
| Prob-L | 92.8 | 94.6 | 99.8 | 0.8 | 91.2 | 71.2 | 97.0 | 91.8 |
| YOLOv11 | 61.0 | 86.4 | 38.2 | 98.4 | 87.0 | 42.0 | 78.6 | 70.4 |
| Ensemble | 99.8 | 100.0 | 99.4 | 96.2 | 94.8 | 76.4 | 97.4 | 99.0 |

We tested the attack transferability of our RGB-T adversarial clothes to unseen RGB-T detectors, both in the digital and physical worlds. The experimental details are described in App. F. It is worth noting that most of the experiments were conducted under a black-box attack setting, which is more challenging but practically significant for real-world applications. The results of the digital experiments are shown in Tab. 5, and the physical world results are available in App. E.

The results indicate that that our method successfully attacked various RGB-T detectors in both white-box and black-box settings. More importantly, our fusion-stage ensemble method effectively improved the ASRs against unseen RGB-T detectors compared to patterns optimized for a single model. This suggests that we can just use one single clothing pattern to attack unseen RGB-T detectors with different fusion architectures.

### 4.10 DEFENSE METHODS

We evaluated the effectiveness of eight typical defense methods against our attack method. These included five traditional defense techniques: Adversarial Training (Goodfellow et al., 2014), Total Variance Minimization (Agarwal et al., 2021), Bit Squeezing (Xu et al., 2017), JPEG Compression (Guo et al., 2017), and Pixel Mask (Guo et al., 2017), along with three state-of-the-art methods specifically designed for defending object detection attacks: PAD (Jing et al., 2024), NAPGuard (Wu et al., 2024), and Jedi (Tarchoun et al., 2023). The experimental details of these methods are provided in App. G. The results show that, although these methods had some defense effects, the ASRs of our method after defense still achieved at least 70%, further indicating the effectiveness of our attack approach.

## 5 CONCLUSION

This paper presents a novel method to RGB-T physical attacks using adversarial clothing with NORP. We construct 3D RGB-T models for human and adversarial clothing to simulate full-view (0°–360°) RGB-T attacks. NORP is a new adversarial pattern design using distinct visible and thermal materials without overlap, avoiding the light reduction in ORP. To simultaneously optimizing continuous RGB pixels and discrete thermal pixels within NORP, we propose an SDCO method. Through systematic evaluation in both the digital and physical world, our work comprehensively reveals the vulnerabilities of RGB-T detectors across different fusion architectures, viewing angles, and distances, which is important for building more robust detectors in the future.

**Ethics Statement:** Adversarial example techniques should be used carefully. If abused, adversarial attacks may threaten the security of AI systems. However, adversarial attacks also advance AI robustness research by exposing system vulnerabilities and promote the development of more robust and trustworthy AI models.

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

## A  DETAILS FOR BUILDING 3D RGB-T MODELS

To construct a 3D RGB-T model, we develop a method for extending an 3D RGB model into an aligned 3D RGB-T model based on a previous thermal 3D modeling approach (Zhu et al., 2024). Initially, we utilized publicly available 3D RGB human and clothing models (Hu et al., 2023) as the foundation. Next, we take a clothing model (Fig. 6) as an example to illustrate our method.

The challenge in building an aligned 3D RGB-T model lies in generating an thermal "skin" that aligns with the 3D mesh model (Fig. 6(c)). To address this, we first unfold the faces of the 3D mesh model into a 2D faces map (Fig. 6(d)) and organize it into different regions, such as the back and arms, using Maya software. Next, we capture real thermal images of clothing using an thermal camera and process them to align with the faces map, producing an aligned thermal texture map (Fig. 6(f)). This process ensures that the real thermal texture is properly aligned with the 3D mesh model, and the final rendered 3D RGB-T models are shown in Fig. 6(a) and 6(e).

We observe that cropped thermal images may not perfectly align with the faces map. To address this issue, we utilize Photoshop's distortion function to fine-tune the cropped thermal images for better alignment. In some cases, the captured thermal images may only contain partial regions corresponding to the faces map. When this occurs, we capture images from different angles and stitch them together to create a complete thermal texture map.

Using the above approach, we obtain a fully aligned thermal texture image for the 3D RGB-T model. Finally, we employ PyTorch3D renderer to map both the RGB texture and thermal texture onto the 3D mesh surface, resulting in a 3D RGB-T model.

We further observed that the thermal characteristics of clothing vary with time and location. To simulate these changing thermal properties, we captured thermal images of the same clothing at different times (day and night) and in different locations (indoor and outdoor). Using the method described above, we constructed 3D thermal clothing models for various scenarios, as shown in Fig. 7. During both the optimization and testing processes, we randomly switch the texture maps of the 3D thermal clothing model to simulate the thermal characteristics of the clothing under different times and locations.

We captured 10 sets of human thermal textures using an thermal imaging camera FLIR T560, covering various environments such as indoor and outdoor settings during both day and night. During the 3D rendering process, visible light is randomly selected from point light sources, directional light sources, and ambient light sources. Infrared light is fixed as an ambient light source to simulate the thermal radiation emitted by the human body in all directions. We increased the brightness of the ambient light source in the thermal rendering to make it close to the thermal imaging effect caused by the high temperature of the human body.

For each sample, we randomly selected the azimuth from 0 to 360 degrees, the elevation from -10 to 20 degrees, and the distance scale from 1.5 to 3.5 for RGB-T joint rendering. After rendering, the human figures were randomly placed within the aligned FLIR background with $x \sim U(-1, 1)$

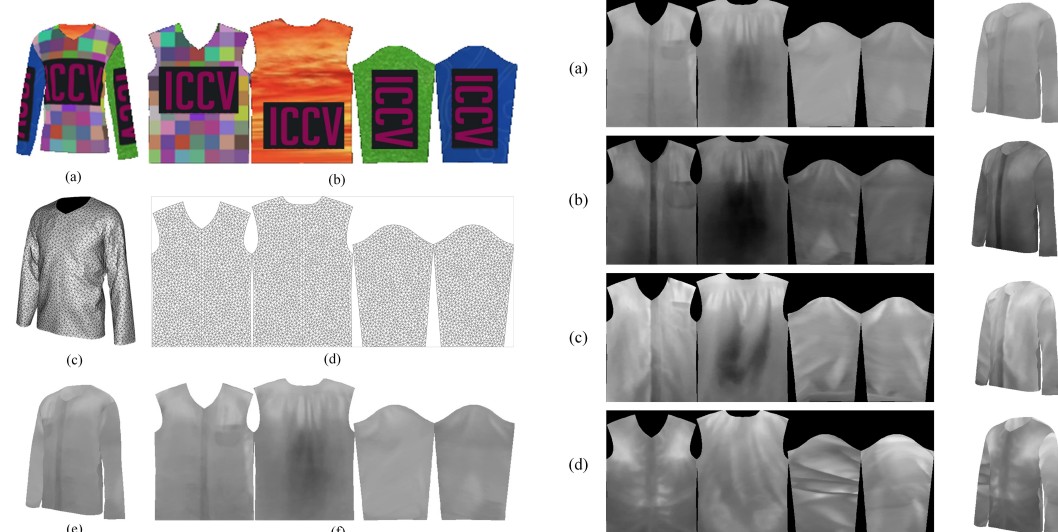

Figure 6: Construction of 3D RGB-T model. (a) 3D RGB model. (b) RGB texture. (c) 3D mesh. (d) Reorganized faces map. (e) 3D thermal model. (f) Thermal texture collected from real world.

Figure 7: Different thermal characteristics of clothes at (a) day indoors, (b) day outdoors, (c) night indoors, and (d) night outdoors.

and $y \sim U(-0.2, 0.2)$, followed by object detection. Considering that some images in the FLIR dataset contain other individuals, their detection results need to be excluded. Therefore, we set an Intersection over Union (IoU) threshold of 0.6 to exclude detection boxes that fall outside the placement range of our rendered human figures, and then took the highest detection score as the loss value.

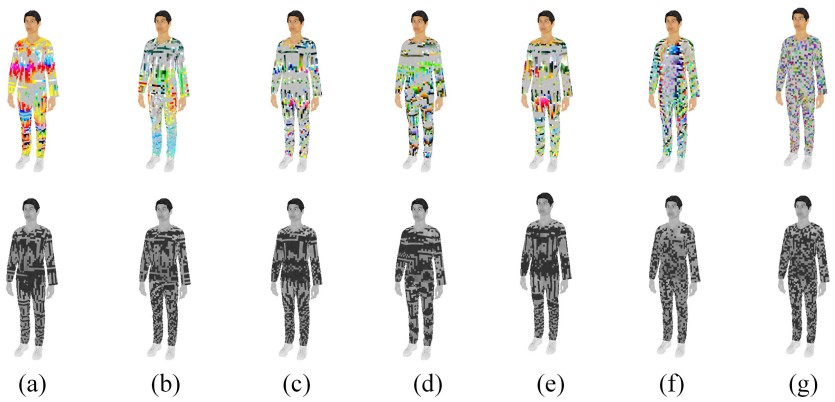

(a)     (b)     (c)     (d)     (e)     (f)     (g)

Figure 8: 3D RGB-T models for (a) Prob-E, (b) Prob-M, (c) Prob-L, (d) Ensemble Detector, (e) YOLOv11, (f) Deformable DETR, and (g) Random RGB-T Pattern.

## B  DETAILS FOR EXPERIMENTAL SETTINGS OF DIGITAL ATTACKS

When conducting spatial discrete-continuous optimization (SDCO), we set the learning rate to 0.01, the batch size to 2, and initialized the attack with $[r_i^{(V)}, g_i^{(V)}, b_i^{(V)}] = U(0,1)$, $\tilde{p}_i = U(0.5 - 0.01, 0.5 + 0.01)$. The number of training steps was set to 10k steps for the Prob-E, Prob-M, and Prob-L models, and 20k steps for the YOLOv11 model. We set the $\alpha$ of the SRD to 0.7 and the pixel size to $10 \times 10$. The texture of shirt was divided into $86 \times 34$ pixels, and the texture of pants was divided into $70 \times 48$ pixels.

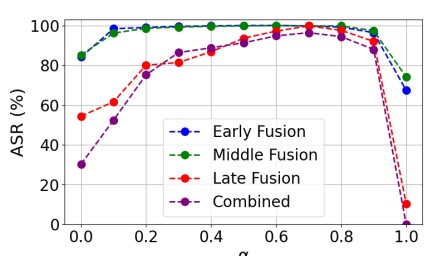

Figure 9: Comparison of different RGB-T attack methods targeting (a) Prob-E, (b) Prob-M, (c) Prob-L, (d) YOLOv11 across diverse scenes in the digital world.

All experiments were performed on RTX3090 GPU with Ubuntu 22.04, CUDA 11.8 and Pytorch 1.13. A single experiment requires around 20GB of GPU memory, takes about 5 hours for every 10k training steps.

After optimization, we rendered the adversarial textures onto the 3D human models (Fig. 8) and placed them in the background images of the test set. We then input the images to RGB-T detectors and evaluate the attack performance both in the white-box and black-box setting. We also compare our method with simple baselines and previous RGB-T attack methods. The results are shown in Tab. 1 and Tab. 5. Fig. 9 shows a set of visual examples.

The results indicates that our method effectively attacked RGB-T detectors with different fusion architectures in the digital world, outperforming simple baselines and previous RGB-T attack methods.

## C  ANALYSIS OF THE KEY PARAMETER OF SRD

We further analyzed the impact of the key parameter of SRD, mask probability $\alpha$, on the ASR. The results are shown in Fig. 10. We observed that highest average ASR was achieved when $\alpha$ was set to 0.7. Therefore, we set $\alpha = 0.7$ in our experiments.

It is worth noting that, $\alpha$ serves as a balancing factor between the optimization parameters of the visible and thermal modalities. When $\alpha$ is too low, the number of trainable parameters in the thermal modality significantly exceeds that in the visible modality, causing the SDCO algorithm to focus more on optimizing the thermal modality than the visible modality. Conversely, when $\alpha$ is too high, SDCO focuses more on optimizing the visible modality. This imbalance in optimization between the two modalities can ultimately degrade the overall attack effectiveness.

Figure 10: Effect of the parameter $\alpha$

## D  EFFECT OF PIXEL SIZE

Another hyperparameter that affects the ASR is the pixel size of the adversarial pattern. We tested the ASRs of adversarial clothing patterns with pixel sizes of 5×5, 10×10, 15×15, and 20×20, targeting RGB-T detectors with different fusion architecture. The results are shown in Tab. 6. We found that adversarial clothing patterns with a pixel size of 10×10 achieved the best performance among these.

## E  MORE EXAMPLES FOR RGB-T PHYSICAL ATTACKS

We provide photos of our adversarial RGB-T clothes in Fig. 12 and additional examples of RGB-T physical attacks in Fig. 11. The captured scenes include both indoor and outdoor environments,

Table 6: Effect of pixel size from 5x5 to 20x20. Note that highest ASR values are marked in **bold**.

| Method | ASR (%) ↑ | | | | |
|---|---|---|---|---|---|
| | Early | Mid | Late | Combined | |
| | | | | RGB | $T$ |
| 5×5 | 99.8 | 99.6 | 99.2 | 86.4 | 85.6 |
| 10×10 | **100.0** | **100.0** | **99.8** | **98.4** | **98.2** |
| 15×15 | **100.0** | 99.8 | 99.0 | 94.2 | 94.6 |
| 20×20 | 99.8 | 99.8 | 97.0 | 87.0 | 88.4 |

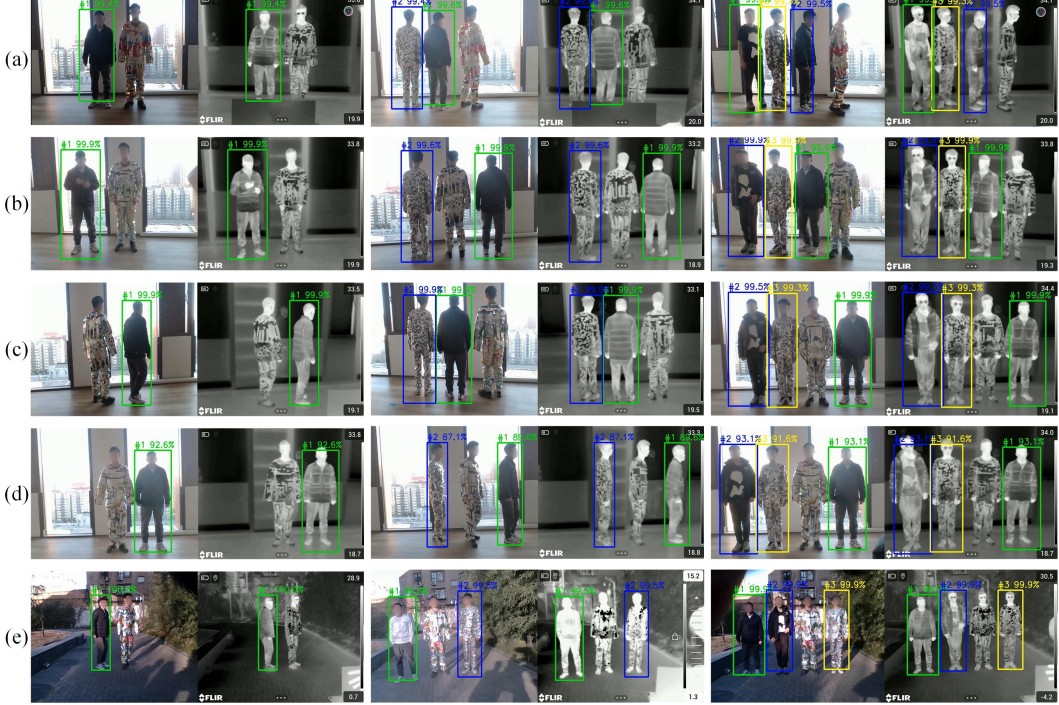

Figure 11: Visualization of physical RGB-T attacks targeting (a) Prob-E, (b) Prob-M, (c) Prob-L, (d) YOLOv11, and (e) Ensemble RGB-T detector at different angles, distances, time, and locations. Facial areas are blurred for privacy reasons.

spanning different times of the day, including morning, noon, afternoon, and nightfall. Volunteers participated in groups of two, three, or four, wearing our RGB-T adversarial clothes (Fig. 12(a-f)), random-pattern RGB-T clothing(Fig. 12(g)), ordinary clothing, or holding a UAP patch.

The tested RGB-T detectors include both white-box models, such as an early-fusion detector Prob-E (Chen et al., 2022), a mid-fusion detector Prob-M (Chen et al., 2022), a late-fusion detector Prob-L (Chen et al., 2022), and independent RGB-T detectors YOLOv11(RGB) and YOLOv11(T) (Khanam & Hussain, 2024), as well as unseen black-box models, including an unseen early-fusion model

Table 7: Transferability in the physical world. The numbers are ASRs.

| Train \ Test | Prob-E | Prob-M | Prob-L | YOLOv11 | RPN-E | AR-CNN | RPN-L | D-DETR |
|---|---|---|---|---|---|---|---|---|
| Prob-E | 83.6 | 71.4 | 21.3 | 1.9 | 47.3 | 51.1 | 48.1 | 47.5 |
| Prob-M | 57.8 | 87.8 | 13.4 | 3.4 | 61.4 | 47.9 | 58.2 | 48.1 |
| Prob-L | 61.6 | 77.4 | 78.4 | 0.3 | 75.1 | 46.2 | 65.2 | 41.2 |
| YOLOv11 | 50.5 | 55.3 | 32.3 | 66.9 | 39.7 | 33.9 | 55.3 | 27.4 |
| Ensemble | 77.6 | 84.3 | 78.2 | 68.8 | 71.4 | 55.6 | 59.0 | 50.0 |

Table 8: Evaluation of defense methods

| Method | ASR drop (%) ↓ | | | | |
| --- | --- | --- | --- | --- | --- |
| | **Early** | **Mid** | **Late** | **Combined** | |
| | | | | RGB | $T$ |
| AT | 23.4 | 18.4 | 21.0 | 25.4 | 24.6 |
| TVM | 17.0 | 14.6 | 9.8 | 14.6 | 18.0 |
| BIT | 9.8 | 7.2 | 8.6 | 12.4 | 14.2 |
| JPEG | 11.2 | 10.0 | 15.4 | 16.8 | 19.6 |
| PM | 21.0 | 17.4 | 24.2 | 12.0 | 10.2 |
| PAD | 17.2 | 12.4 | 21.2 | 25.0 | 22.0 |
| NAPGuard | 15.4 | 11.0 | 22.4 | 28.6 | 17.2 |
| Jedi | 22.4 | 19.8 | 29.8 | 27.6 | 28.2 |

RPN-E(Liu et al., 2016), an unseen mid-fusion model AR-CNN(Zhang et al., 2019), an unseen late-fusion model RPN-L(Liu et al., 2016), and unseen independent RGB-T detectors D-DETR(RGB) and D-DETR(T)(Zhu et al., 2020). The camera angles cover a full 0-360° range, and the capture distances range from 2 to 15 meters.

Experimental results indicate that our adversarial clothing successfully evades multiple white-box and black-box RGB-T detectors across various scenes, angles, and distances, consistently outperforming the baseline methods.

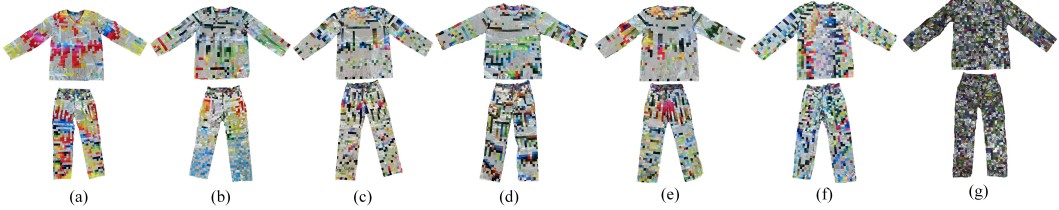

Figure 12: Physical RGB-T clothes for (a) Prob-E, (b) Prob-M, (c) Prob-L, (d) Ensemble Detector, (e) YOLOv11, (f) Deformable DETR, and, (g) Random Pattern.

## F    ATTACK TRANSFERABILITY IN THE PHYSICAL WORLD

We tested the attack transferability of our RGB-T adversarial clothes to unseen RGB-T detectors in the physical world. The RGB-T detectors involved in the optimization process include an early-fusion detector Prob-E (Chen et al., 2022), a mid-fusion detector Prob-M (Chen et al., 2022), a late-fusion detector Prob-L (Chen et al., 2022), and independent RGB-T detectors YOLOv11(RGB) and YOLOv11(T) (Khanam & Hussain, 2024), and an ensemble model of the aforementioned models. In the testing process, the models, in addition to the ones listed above, also include an unseen early-fusion model RPN-E(Liu et al., 2016), an unseen mid-fusion model AR-CNN(Zhang et al., 2019), an unseen late-fusion model RPN-L(Liu et al., 2016), and unseen independent RGB-T detectors D-DETR(RGB) and D-DETR(T)(Zhu et al., 2020). The results of the physical experiments are shown in Tab. 7. Please note that we calculated the average ASR of both modalities for independent RGB-T detectors in Tab. 7.

The results indicate that our method can transfer to a variety of unseen RGB-T detectors in the physical world. More importantly, our fusion stage ensemble method effectively improves the ASRs against unseen RGB-T detectors compared to patterns optimized for a single model. This suggests that we can just use one single clothing to attack unseen RGB-T detectors with different fusion architectures in the physical world.

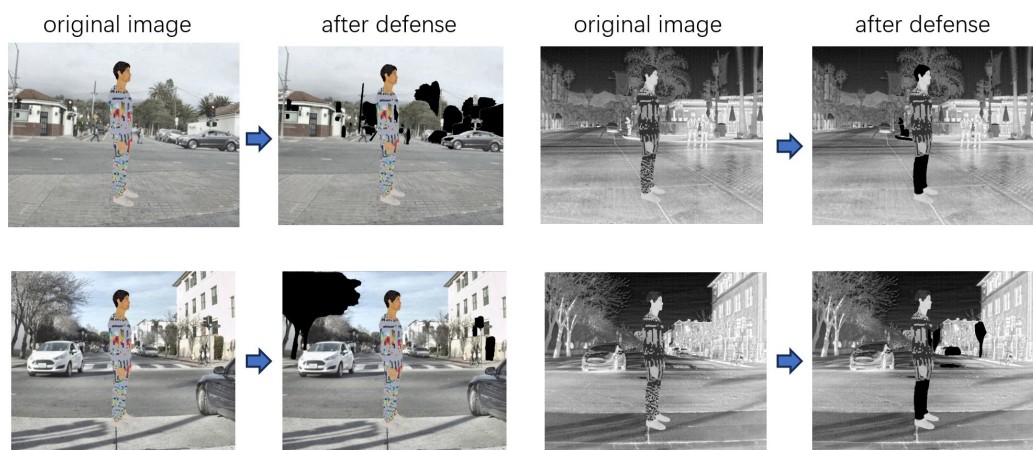

Figure 13: Failure cases of defense. Black segments denote adversarial regions identified by defense algorithms.

Table 9: Evaluation on E2E-MFD and DAMSDET.

| Method | ASR (%) ↑ | |
|--------|-----------|---------|
| | E2E-MFD | DAMSDET |
| Clean | 0.2 | 3.6 |
| Random | 0.4 | 9.6 |
| MAP | 8.8 | 17.8 |
| MIC | 11.4 | 14.2 |
| UAP | 7.0 | 12.0 |
| Ours | 88.2 | 94.6 |

## G  DETAILS FOR DEFENSE METHODS

We evaluated the effectiveness of eight typical adversarial defense methods against our attack methods. These included five traditional defense techniques: Adversarial Training(Goodfellow et al., 2014), Total Variance Minimization (Agarwal et al., 2021), Bit Squeezing(Xu et al., 2017), JPEG Compression(Guo et al., 2017), and Pixel Mask(Guo et al., 2017), along with three state-of-the-art methods specifically designed for defending against adversarial attacks on object detectors: PAD(Jing et al., 2024), NAPGuard(Wu et al., 2024), and Jedi(Tarchoun et al., 2023).

For Adversarial Training, we began by rendering 200 images with adversarial textures and collecting their corresponding labels. These images were then added to the original FLIR-aligned dataset for fine-tuning RGB-T detectors. We fine-tuned the Prob-E, Prob-M, Prob-L, and YOLOv11 models individually for 10 epochs, with a learning rate of 0.001. The fine-tuned model weights were subsequently used as the target for our attacks. In the case of Pixel Mask defense, we randomly selected an $80 \times 80$ region on both the shirt and trousers textures, simultaneously removing the thermal materials and visible light adversarial textures from these areas. For Bit Squeezing, we reduced the bit depth of both the 8-bit visible light and thermal adversarial textures to 7-bit. Regarding Total Variance Minimization and JPEG Compression, we utilized modules from the Adversarial Robustness Toolbox library to compress or blur the adversarial textures. As for PAD, NAPGuard, and Jedi, we directly implemented their open-source codes, which attempt to verify and eliminate our adversarial clothing on simulated humans.

Tab. 8 shows the results. It indicates that although these methods had some defense effects, the ASRs of our method after defense still achieved at least 70%, further indicating the effectiveness of our attack approach. Note that our 3D modeling ensures that adversarial clothing can cover larger areas of human body and have more irregular shapes and boundaries compared with 2D adversarial patches. Therefore, even latest methods specialized in defending object detectors cannot

precisely locate our adversarial clothing. Fig. 13 illustrates examples of failure cases of defense when applying PAD method.

## H    ATTACK DEFORMABLE DETR

To evaluate the attack effectiveness of our method against the transformer-based detector, we attacked a typical detector, Deformable DETR (Zhu et al., 2020), both in the digital and physical world. Following the experimental setup described in Sec. 4.4, we obtained the optimized adversarial clothing, as shown in Fig. 8(f). We then tested its attack performance in the digital world, and the experimental settings were same as Sec. 4.4. The ASR of the adversarial clothing was 99.6% in the digital world. In comparison, the ASR of the random pattern clothing was only 17.8%.

Next, we manufactured adversarial clothes based on the optimized patterns, as shown in Fig. 12(f), and conducted physical experiments following the setup described in Sec. 4.9. The physical experiments indicated that our adversarial clothes successfully evaded Deformable DETR in the physical world, achieving an ASR of 75.4%, while the ASR of random pattern clothes was only 22.0%. These results, together with our previous experiments, indicate that our method is effective against both the CNN-based models (e.g., YOLOv11) and the transformer-based model, highlighting the generality of our approach.

## I    ATTACK E2E-MFD AND DAMSDET

We evaluated our attack method on two recently published RGB-T detectors—an early-fusion detector E2E-MFD (Zhang et al., 2024) and a mid-fusion detector DAMSDET(Guo et al., 2024). For a fair comparison, we employed the clean clothing pattern (pure color pattern), random RGB-T pattern (without optimization), and the adversarial patterns generated by previous works including MAP(Kim et al., 2022), UAP(Wei et al., 2023b), and MIC(Kim et al., 2023). The results are shown in Tab. 9. Our method achieved an average ASR of 91.4%, while the ASR for the control group was below 17.8%. This indicates that our approach can effectively attack state-of-the-art RGB-T detectors, outperforming simple baselines and previous RGB-T attack methods.

## J    LIMITATION AND FUTURE WORK

As the first physical attack that effectively targets RGB–T detectors across all fusion methods, this paper accepts a modest loss in garment naturalness, consistent with early work on single-modality (Kim et al., 2022; 2023; Zhu et al., 2021; 2022; Hu et al., 2022), to give research priority on better exposing the vulnerabilities of current RGB–T object detectors. Future work would further focus on how to improve the perceptual naturalness of clothing patterns while remaining effectiveness of attack in multi-modal real-world detection.

