# OpenReview forum: "Multimodal Physical Adversarial Clothing Evades Visible-Thermal Detectors with Non-Overlapping RGB-T Pattern"
_ICLR.cc/2026/Conference — ICLR 2026 Conference Withdrawn Submission_

### Official Review · Reviewer_UtUf · 2025-10-28

**Soundness:** 3
**Presentation:** 3
**Contribution:** 3
**Rating:** 4
**Confidence:** 5

**Summary:**

This paper addresses the overlooked physical-world security of RGB-T detectors by proposing adversarial clothing based on a Non-Overlapping RGB-T Pattern. It constructs 3D RGB-T models of the human body and adversarial clothing to enable full-view (0°–360°) attacks, and designs a Spatial Discrete-Continuous Optimization (SDCO) method to solve the collaborative optimization challenge between continuous RGB pixels (visible modality) and discrete thermal pixels (thermal modality) in NORP. Experiments show that the method achieves a good attack effectiveness in the physical world against RGB-T detectors with different fusion architectures. Additionally, the proposed fusion-stage ensemble method significantly enhances the transferability of adversarial attacks to unseen detectors, comprehensively revealing the vulnerabilities of RGB-T detectors across varying viewing angles, distances, and fusion architectures.

**Strengths:**

1. The paper covers diverse test scenarios, including full viewing angles (0°–360°), multiple distances (2–20m), indoor/outdoor environments, and day/night timeframes.
2. It focuses on physical adversarial attacks against RGB-T detectors, filling the gap in practical security evaluation for scenarios such as CCTV surveillance.
3. The proposed Spatial Discrete-Continuous Optimization effectively resolves the coordination problem between continuous RGB pixels (visible modality) and discrete thermal pixels (thermal modality), ensuring the physical realizability of the adversarial pattern.

**Weaknesses:**

1. The proposed SDCO method relies heavily on empirically determined optimal hyperparameters (e.g., mask probability $\alpha$=0.7, pixel size 10×10), but lacks exploration of these parameters’ performance across different scenarios.
2. How about the optimization stability? As the core contribution of this paper, the SDCO algorithm would benefit from presenting training loss curves to better demonstrate the method’s effectiveness.
3. The paper lacks detailed manufacturing guidelines, making it difficult for other researchers to replicate the exact physical artifacts.
4. Writing issue: An extra line break exists at the end of Section 2.1.

**Questions:**

1. The human images in Fig. 2, Fig. 8, and Fig. 9 are from the same instance. Additionally, the human subjects and environmental backgrounds have a significant domain gap from the real world. How to address the simulator-to-real-world domain gap?
2. What is the confidence threshold used in real-world experiments? Is it equal to 0.6 as described in Section 4.3?
3. Why is UAP’s attack effectiveness lower than that of the Random pattern? This result is counterintuitive.

---

### Official Review · Reviewer_B9rf · 2025-10-30

**Soundness:** 3
**Presentation:** 2
**Contribution:** 3
**Rating:** 4
**Confidence:** 5

**Summary:**

- The paper proposes NORP, a non-overlapping RGB-Thermal pattern for adversarial clothing that combines printed RGB fabric with aluminum film, optimized jointly via a Spatial Discrete-Continuous Optimization (SDCO) scheme. On four white-box RGB-T detectors spanning early/mid/late fusion and independent detectors, NORP attains ~100% digital ASR and ~61–79% physical ASR; SDCO also beats STE and Gumbel-Softmax. The paper evaluates transfer and several defenses.

**Strengths:**

- The work tackles an important, under-explored multimodal physical threat model and proposes a method (NORP + SDCO) that is conceptually neat, physically grounded, and empirically strong across fusion architectures, with real garments.

**Weaknesses:**

- While NORP is compared against MAP, MIC, and UAP, these baselines are relatively dated. The evaluation omits newer multimodal and RGB-only physical attacks that employ differentiable rendering, optimization under lighting/EOT constraints, or diffusion-based perturbations (e.g., Adv3D, PRoCA, DiffusionPatch). Without benchmarking against these, it is unclear whether NORP’s improvements stem from its novel discretization strategy or simply from stronger optimization.
- The paper acknowledges visual realism but offers limited quantitative or human-subjective evaluation. Since clothing-based attacks must balance stealth with adversarial efficacy, user studies or perceptual metrics (LPIPS, FID, or texture-similarity) would help verify that NORP remains inconspicuous to humans. Current qualitative figures are promising but insufficient for claims of “naturalness”.
- The evaluation defines attack success using IoU = 0.5 and confidence ≥ 0.6. This relatively strict threshold may exaggerate the apparent success rate, especially since modern detectors often report low-confidence bounding boxes that still indicate detection. A sensitivity analysis over thresholds (e.g., 0.3–0.7) would show how robust the reported ASR is to this design choice.

**Questions:**

Please check weaknesses

---

### Official Review · Reviewer_qGd2 · 2025-10-31

**Soundness:** 3
**Presentation:** 3
**Contribution:** 3
**Rating:** 4
**Confidence:** 3

**Summary:**

This paper proposes a physical adversarial clothing method to evade RGB-T object detectors. By designing a NORP and using 3D modeling for full-view attack simulation, the approach jointly optimizes visible and thermal perturbations via a Spatial Discrete-Continuous Optimization     method. Experiments show high attack success rates in both digital and physical settings against detectors with various fusion strategies.

**Strengths:**

1.Establishes the first practical full-view physical attack against multimodal detectors through comprehensive 3D RGB-T modeling and rendering.

2.Introduces an innovative NORP design that effectively coordinates visible and thermal perturbations using inexpensive, readily available materials.

3.Demonstrates remarkable cross-architecture effectiveness with high attack success rates against various fusion strategies in both digital and physical environments.

**Weaknesses:**

1.Limited evaluation under extreme environmental conditions such as heavy rain, snow, fog, or rapidly changing thermal scenarios.

2.Dependence on Specific Optimization: The SDCO method relies heavily on the precise optimization of patterns, which may be challenging to replicate in different environments or for other detection systems.

3.Compromised stealth and naturalness of clothing patterns, potentially raising suspicion in real-world deployment.

**Questions:**

1.How can the perceptual naturalness of the adversarial clothing be improved while maintaining high attack effectiveness?

2.When under challenging but common environmental conditions, such as rain, snow, and intense sunlight, how would the adversarial clothing perform , which could significantly alter its thermal signature?

3.Generalization to Non-Human Targets: The method is designed and evaluated specifically for evading pedestrian detectors. How effectively could this approach be adapted to create adversarial patterns for other critical objects in autonomous driving, such as vehicles or cyclists, which have vastly different shapes and thermal signatures?

---

### Official Review · Reviewer_RaUJ · 2025-11-12

**Soundness:** 2
**Presentation:** 2
**Contribution:** 2
**Rating:** 2
**Confidence:** 4

**Summary:**

This paper studies physical-world adversarial attacks on visible thermal RGB T detectors by using non overlapping RGB T patterns on clothing. The authors construct 3D RGB T models and propose the NORP pattern together with a spatial discrete continuous optimization method to design adversarial garments. Experiments report high attack success in both digital and physical tests and a fusion stage ensemble that improves transferability across detectors.

**Strengths:**

- The topic of security in RGB-T detector is interesting. The idea of using adversarial clothing is also interesting.
- The paper is easy to follow.
- The results demonstrate the effectiveness of the proposed method.

**Weaknesses:**

- The abstract should be improved. More information is needed to understand the abstract. For example, it is not clear why NORP can avoid the light reduction in overlapping RGB-T patterns (ORP).
- Fig. 2 is not very comprehensive as a main figure to show the pipeline. More information could be provided in the figure.
- It is not clear whether the proposed method can be extended to other RGB-T models in addtion to RGB-T detectors.
- Misalignment in RGB-T datasets is common. It is not clear whether misalignment between RGB and thermal images will affect the performance of the proposed method. This paper also uses fusion at different levels. What are the effects of misalignment on different levels of fusion?
- The target RGB-T detectors presented in Section 4.2 are outdated. More latest RGB-T detectors should be used in performance evaluation.
- In Section 4.3, the authors mentioned target pedestrians. What are target pedestrians?
- Only one evaluation metric is utilized in this paper.

**Questions:**

- The abstract should be improved to make is easier to understand.
- Fig. 2 should be improved.
- Can the proposed method be applied to other RGB-T models?
- It is not clear whether misalignment between RGB and thermal images will affect the performance of the proposed method. This paper also uses fusion at different levels. What are the effects of misalignment on different levels of fusion? More details should be provided.

---

### Note · Authors · 2025-11-12

I have read and agree with the venue's withdrawal policy on behalf of myself and my co-authors.